# Extracellular Vesicles as Drug Delivery Systems in Cancer

**DOI:** 10.3390/pharmaceutics12121146

**Published:** 2020-11-26

**Authors:** Laia Hernandez-Oller, Joaquin Seras-Franzoso, Fernanda Andrade, Diana Rafael, Ibane Abasolo, Petra Gener, Simo Schwartz

**Affiliations:** 1Drug Delivery and Targeting Group, Molecular Biology and Biochemistry Research Centre for Nanomedicine (CIBBIM-Nanomedicine), Vall d’Hebron Institut de Recerca, Universitat Autònoma de Barcelona, 08035 Barcelona, Spain; laia.hernandez@alumni.vhir.org (L.H.-O.); joaquin.seras@vhir.org (J.S.-F.); fernanda.silva@vhir.org (F.A.); diana.fernandes_de_so@vhir.org (D.R.); ibane.abasolo@vhir.org (I.A.); 2Networking Research Centre for Bioengineering, Biomaterials, and Nanomedicine (CIBER-BBN), Instituto de Salud Carlos III, 50004 Zaragoza, Spain

**Keywords:** cancer stem cells, extracellular vesicles, drug delivery systems

## Abstract

Within tumors, Cancer Stem Cell (CSC) subpopulation has an important role in maintaining growth and dissemination while preserving high resistance against current treatments. It has been shown that, when CSCs are eliminated, the surrounding Differentiated Cancer Cells (DCCs) may reverse their phenotype and gain CSC-like features to preserve tumor progression and ensure tumor survival. This strongly suggests the existence of paracrine communication within tumor cells. It is evidenced that the molecular crosstalk is at least partly mediated by Extracellular Vesicles (EVs), which are cell-derived membranous nanoparticles that contain and transport complex molecules that can affect and modify the biological behavior of distal cells and their molecular background. This ability of directional transport of small molecules prospects EVs as natural Drug Delivery Systems (DDS). EVs present inherent homing abilities and are less immunogenic than synthetic nanoparticles, in general. Currently, strong efforts are focused into the development and improvement of EV-based DDS. Even though EV-DDS have already reached early phases in clinical trials, their clinical application is still far from commercialization since protocols for EVs loading, modification and isolation need to be standardized for large-scale production. Here, we summarized recent knowledge regarding the use of EVs as natural DDS against CSCs and cancer resistance.

## 1. Introduction

Despite achieving great advances in oncology in the past few years, in terms of treatment and patient survival, cancer still represents the second cause of death worldwide. In particular, treatment resistance and metastasis in vital organs account for 90% of cancer related deaths [1]. New treatments and therapeutic approaches are needed to successfully fight tumor resistance, cancer progression and metastasis to improve clinical outcomes.

A tumor is a highly complex and heterogenic dynamic entity that evolves over time, adapting and therefore surviving to adverse conditions [2]. As the disease progresses, it becomes more difficult to treat, since it spreads to distant organs and/or acquires resistance to the treatment [3,4,5].

Within the tumor, a heterogeneous mix of different environments and cell types, such as CSCs (Cancer Stem Cells), DCCs (Differentiated Cancer Cells), CAFs (Cancer Associated Fibroblasts), mesenchymal cells, tumor-infiltrated immune cells, endothelial cells and stromal cells can be found. All of them are located within the extracellular matrix and together contribute to disease progression [6,7,8,9,10]. The constant exchange of information among these cells is essential to guarantee survival and progression of the tumor and to orchestrate the coordination and collaboration of different cells.

In this review, we assess the versatility of TME (Tumor Micro Environment) and the main player of communication within TME, the Extracellular Vesicles (EVs) and its use as a drug delivery system (DDS) for cancer treatment. We cover mostly in vitro studies taking into account the main cells currently used as EV sources. However, in vivo studies and a few ongoing (phase I or II) clinical studies are also described. We advocate for EVs, either natural or engineered, by comparison with liposomes or synthetic drug delivery particles. Indeed, we are well aware of the many challenges, which remain to be solved in order to translate the worldwide increasing current knowledge about EVs from the bench to the clinical care of cancer patients.

### Cancer Stem Cells, Cancer Resistance and Cell Communication

CSCs substantially contribute to tumor growth and progression. They are an undifferentiated subset of cells within the tumor with stem-like properties, high proliferation rate, ability to differentiate and self-renewal potential [2,11,12,13,14]. CSCs are believed to sustain uncontrolled tumor growth and to be responsible for cancer progression, recurrence, metastatic spread, invasiveness, multidrug resistance and treatment failure [4,6,7,15,16,17,18]. Therefore, after treatment, the tumor percentage of CSC subpopulation frequently rises when compared to other tumor cells types [4,5,6,19]. It has been shown that only a few CSCs are needed for tumor regeneration in vivo, and they can enter to an undetectable quiescence state when the conditions of the TME are not favorable and proliferate afterwards [2,4,17].

Until the 1990s, the initiation and progression of a tumor was explained by the clonal cancer model, in which cancer was thought to be driven by accumulated somatic mutations that confer uncontrolled growth, a more aggressive behavior and higher fitness to a malignant transformed cell [17,20].

However, it was later shown that not all the cells within the tumor presented the same tumorigenic potential. With this knowledge, a hierarchical model (also referred to as the CSC model) (Figure 1A) has been described. Accordingly, only a small and distinct subpopulation of CSCs is alleged to have the capacity to generate and maintain the tumor [4,5]. In this model, cancer cells are created from a precursor cell, which undergoes either symmetric (generating two CSCs or two DCCs) or asymmetric (generating a CSCs and a DCCs) divisions [10]. Here, DCCs do not present the ability to self-renew indefinitely and can only generate cells of their same type. On the other hand, CSCs can generate multiple and heterogeneous tumor subpopulations that differentiate into diverse lineages [17,21]. In terms of cancer treatment, according to the hierarchical model, the complete eradication of the CSCs population should be enough to eradicate the tumor and prevent the relapse of the disease [6,13]. Therefore, strong efforts have been invested over the past decade, in the identification of CSCs within the tumor in order to target treatments against them [10].

Nevertheless, this hierarchical model cannot explain the dynamic behavior seen in the CSC subpopulation, as the concepts of DCC and CSC were not conceived within the same cell [16]. Therefore, a new stochastic model has been recently postulated (Figure 1B) [16,22,23]. According to this model, a tumor is composed by different cell populations that maintain a stable communication among them. Through this communication, they can “sense” if one specific subpopulation of the tumor cells is redundant, absent or has been depleted. In this model, the amount of CSCs seems to remain constant, to maintain the mentioned equilibrium within the TME [4,6,17]. According to a stochastic model, any cell of the tumor can initiate the progression of the disease due to the existing phenotypic plasticity, and, further, any cancer cell can recover the stem-cell-like phenotype by dedifferentiation [6,13,24].

Although the hierarchical model and the stochastic model have different considerations regarding the importance of CSCs in tumor initiation and progression, they are not mutually exclusive because of cellular plasticity (Figure 1) [6]. Essentially, the tumor is formed in a hierarchical manner, that is unstable since constant stochastic actions allow for the introduction of newly hierarchically organized cell populations [6,25,26].

In this context, cancer cells interact with other cells from the TME through direct cell–cell contact and/or using paracrine signaling, particularly for distant cells. Both cellular and non-cellular components of the niche have a role in maintaining stable the stemness potential of the tumor and further regulate CSC plasticity and EMT (epithelial to mesenchymal transition) [6,17,27,28].

Besides, different stress situations can also have critical involvement in the initiation and progression of the tumor. Previous studies have reported the important influence of hypoxia, intratumoral pH and other stress conditions, such as chemotherapeutic treatments, in the CSC niche. These stimuli can promote angiogenesis and the activation of stemness genes and therefore can initiate the dedifferentiation of DCCs [17,29]. Moreover, CSCs present adaptations to survive under hypoxia or acid environments [7]. These examples support the existence of a controlled balance between both cellular populations within the TME and suggest that any alteration in their stable state can have a potential influence in the clinical outcome of a patient [17].

Elucidating the molecular mechanisms that govern cellular plasticity may be essential to overcome the challenge that current therapies face when fighting against cancer. Effective targeting therapies need to be developed to eliminate the roots of continuously evolving tumor cell populations and to avoid the regeneration of CSCs [17,25].

## 2. Extracellular Vesicles in Cellular Communication

As mentioned before, CSC are not a static cell subpopulation of tumor cells, but a population with a highly dynamic phenotype [12]. However, the mechanisms behind these phenomena are still unclear. Recent evidence suggests that the molecular crosstalk between CSCs and DCCs within the TME has a determining role in this process [12]. Moreover, as the reversion process seems to be an important factor for the tumor to gain therapeutic resistance, this crosstalk may represent a crucial mechanism to promote tumor survival. Extracellular vesicles (EVs), mostly exosomes, derived from CSCs are probably one of the most important elements of this crosstalk (Figure 2a) [12,30].

EVs are naturally cell-derived membranous nanoparticles that contain and transport a wide range of complex molecules such as proteins, nucleic acids, sugars and lipids to specific targeted cells, which can affect and modify their behavior [31,32,33,34,35]. They form an endogenous natural transport system throughout which biomolecules can be exchanged among neighboring recipient cells or even to distant organs [1]. They carry a similar set of molecules as the original cell, reflecting thus its biological status, which changes in pathological conditions such as cancer [34,36]. Almost all cell types have been reported to release these vesicles, including human cells [33]. EVs are commonly classified conferring to their biosynthesis mechanism and size in three subtypes: exosomes, microvesicles and apoptotic bodies [37].

As introduced previously, EVs have been found to participate in the pathogenesis of cancer cell-to-cell communication, as they can transport biomolecules that promote tumor growth, cancer progression, treatment resistance and facilitate metastasis [1,34,38,39,40].EVs participate in the creation of the pre-metastatic niche and can also influence therapeutic efficacy, as they may grant chemo-protective properties for tumor cells [41,42]. Among EVs, exosomes present inherent roles in cell-to-cell communication because of their small size (30–100 nm in diameter [12]) and the expression on their lipid bilayer membrane of cell-specific markers [43]. Exosomes are considered to be information carriers between CSC and other cells within the TME, which are essential for the survival of the tumor [12,40,44]. Of interest, it has been shown that cancer cells secrete significantly more exosomes than other cells [45]. Those secreted from cancer cells can regulate the cellular metabolism of the recipient cells, reprograming them to promote or enhance functions such as EMT, apoptosis, proliferation, angiogenesis, immune response suppression, stemness and cellular migration, as they may carry and transfer stemness-related molecules, oncogenic factors and capacity of multidrug resistance to antitumoral treatment [1,12,40,46,47,48,49].

Many studies have shown that cancer cell-derived exosomes can also affect and change the surrounding microenvironment by reprogramming the stromal cells to create a favorable niche for tumor progression [10,38,40,41]. At the same time, CSCs can modulate all components of the tumoral niche, facilitating CSCs growth and dissemination [10]. This tumoral niche components can simultaneously regulate the required equilibrium of CSCs and the process of cellular plasticity and are required for the maintenance of the CSC population (Figure 2b) [10,12,17,27].

## 3. Extracellular Vesicles as Natural Drug Delivery Systems

An ideal anticancer therapy should therefore target both CSCs and DCCs, and also the signals that promote the reversion to the CSC phenotype, to avoid progression and future relapses of the disease.

In this context, specific drug delivery may promote efficiency of anti-cancer therapy. The use of synthetic nanoparticles as drug delivery systems has been in the limelight for the past years [50,51]. Nanomedicine uses nanoscale materials, ranging from 1 to 1000 nm of size, as a consequence of their unique medical benefits regarding their structure and functionality [50]. However, rather few results have been reported in cancer drug delivery and only few prototypes have reached clinical trials; thus, very few of these treatment strategies have successfully transited from bench to bedside [26,52,53]. The main challenges of nanocarriers are still inadequate PK/PD (pharmacokinetic/pharmacodynamic) features, toxicity and immunogenicity, and unspecific targeting capacity. In addition, conventional DDS present significant difficulties in overcoming natural barriers and reaching their expected targets. Therefore, a new focus and a new paradigm of this prospective scientific field is urgently needed [33].

In this context, EVs are attractive, promising candidates to optimize drug delivery for clinical uses. These novel carriers present inherent targeting abilities and less immunogenicity than synthetic nanoparticles and seem to be able to successfully deliver drugs to the tumor site. Nevertheless, our current knowledge on the functions of the molecules exposed at the external surface or incorporated in the lumen of the EVs is still very limited, which hampers the exploitation of specific therapeutic and diagnostic uses and their translation into clinical applications. It is necessary to gain insight in the fundamental processes of EVs biology, to understand the basic mechanisms by which these vehicles can load their specific cargos and target specific cell types, as well as orchestrating their different functional roles as intercellular messengers.

Exosomes, which are considered to develop a principal role in cell-to-cell communication in cancer and in the maintenance of the dynamic equilibrium of CSCs, are being investigated for their potential use against cancer [10,12].

EVs are proposed as natural carriers with manipulated cargo to fight multidrug resistance and metastatic dissemination. EVs may well have advantages compared to the currently available DDS, as they seem to be stable in circulation, they can inherently overcome biological barriers (even the blood-brain barrier), and present intrinsic cell-specific targeting properties [1,12,33,54]. Additionally, EVs can avoid phagocytosis, present significantly low autologous immunogenicity and may use endogenous mechanisms for cargo uptake, trafficking and delivery [1,33,54,55]. The structure of the EVs resembles liposomes but with a more complex lipid layer composition. This complexity in the composition of their membrane helps to deliver the carried material directly into the targeted cell [10]. Moreover, EVs surface markers can be modified or replaced to enhance tumor-targeting specificity, and reduce the systemic toxicity [12,34]. For example, EVs could be coated with CSC marker antibodies to direct the vesicle to this specific cell population within the tumor, which is responsible for tumor progression [12]. They could also be used to present tumor CSC-specific antigens to T cells and consequently help the immune system to fight the disease more efficiently. Up today, chemotherapeutic drugs delivered by exosomes have been shown to have much more stability and effectiveness without toxicity, compared to conventional therapies [12].

Besides, the presence in EVs of different cell-type specific molecular signatures as biomarkers has placed them at the forefront of diagnostics in a wide variety of diseases. Currently, there is a huge interest in applying EVs or synthetic EVs as drug delivery systems. However, not all components in the natural EVs are essential for their function and delivery properties [10]. Therefore, understanding which are the crucial components of natural EVs responsible for specific biological functionalities, such as efficient homing to target cells and efficacious intracellular delivery of their cargo, is still a focus of current studies.

Among all properties that make EVs prospective candidates for drug delivery, probably their most interesting quality is their capacity to transmit nucleic acids and proteins to other cells, and the possibility to directly release their cargo into the cytoplasm of the recipient cells [10,33]. For instance, siRNA and microRNA could be delivered to specific organs or tissues with the aim to target CSC signaling pathways or for gene therapy [12]. Besides, it is important to comprehend the role of EVs in cancer progression and metastatic growth in order to use this knowledge against the disease [43].

### 3.1. Sources of EV-Based Drug Delivery

During the production process of EVs for cancer treatment, it is important to ensure optimal consideration of certain variables known to influence EV properties: cell type, cell collection process and/or expansion methods, the triggering mechanism for the release of EVs and the isolation and storage methods. All these steps can affect EVs population size, membrane markers (especially important for targeting), purity and content [43]. Applications for EVs as drug delivery vehicles includes allogenic and autologous treatments [34,56]. Moreover, the cell source of these EVs is important to avoid immune rejection responses and to allow specific applications. Until now, the most used cell sources of EVs for drug delivery have been immune cells, mesenchymal stem cells (MSCs), cancer cells, and commonly used commercial cell lines.

#### 3.1.1. Immune Cell-Derived EVs

Immune cell-derived EVs are especially promising for cancer therapy, as they seem to share a common action mechanism with the cellular function of their secreting cells, and therefore protect against different diseases and foreign antigens. For instance, natural killer cells when fighting cancer cells, secrete EVs containing different cytotoxic proteins, which are targeted to kill those specific cells and stimulate the action of the immune system [34,57,58]. Therefore, genetic engineered EVs derived from the immune system cells, may represent an advantage for cancer treatment. This has already been seen in vitro and in vivo models [59,60,61]. In a recent study, EVs derived from macrophages and loaded through sonication with Paclitaxel (PTX) prevented metastasis in a lung cancer mouse model. Moreover, when these EVs were modified to reduce their immunogenicity by adding an aminoethylanisamide-polyethylene glycol vector, the EVs enhanced their circulation time and were directly targeted to lung metastases [59,60,61].

#### 3.1.2. MSC-Derived EVs

MSC-derived EVs come from a cell source thought to possess limited immunogenicity and consequently, are suitable for allogenic transplantation. This happens when the expression of co-stimulatory molecules, such as class I major histocompatibility complex molecules, is very low. This quality would be a major goal to avoid immune rejection of the treatment. Moreover, EVs from MSCs present inflammatory tropism and one of their natural functions is to exert therapeutic effects, which comes along with the desired purpose of the natural drug delivery systems [34,53,57]. However, until now, MSCs have had limited use in therapy because of their potential oncogenicity. Nevertheless, several studies have been carried out, using MSCs as a source of EVs for different treatments. As, for example, tumor proliferation has been inhibited by PTX-loaded EVs, which had been released from PTX-treated MSCs in vitro [58,62]. Moreover, MSCs-EVs have reached clinical trials in regenerative medicine for tissue repair after myocardial infarction [63]. In this pathology, the affected cardiomyocytes are usually replaced by a non-elastic collagen scar, which impairs the heart function. However, MSC-derived EVs have been demonstrated to improve recovery after myocardial infarction by promoting neoangiogenesis [63].

#### 3.1.3. Cancer Cell-Derived EVs

Cancer cell-derived EVs are produced in large quantities with special homing abilities due to the TME influence [34,57]. EVs produced in cancer cells express tumor-specific antigens on their membrane, which could help in the generation of the anti-tumor immune response, which has been recently confirmed using a mouse model [34,58]. Moreover, it has been seen that EVs from cancer cells loaded with chemotherapeutic agents can reduce resistance of CSCs to the applied treatment [12,34]. For example, a mouse lung cancer model study showed promising results, as when chemotherapy-loaded EVs were injected, the tumor load was reduced and therefore the survival was prolonged when compared to free chemotherapy treatment [34]. On the other hand, a clinical study has used cisplatin-loaded EVs from A549 human lung cancer cells in three end-stage lung cancer patients, resistant to cisplatin. The results showed that the global quantity of tumor cells and the incidence of CSCs was reduced drastically, while treatment with free cisplatin did not show any beneficial effects for the patients [34,64].

However, it is very important to take into consideration the possibility that this type of EV could also cause tumor growth and metastasis, since such EVs may help tumors to adapt and survive, particularly by the activation of pathological pathways and exerting immune-suppressive effects. Consequent studies were performed to assess how to block this immune-suppressive response, and it was found that, when those EVs were mixed with the adequate immune stimulatory adjuvants, the immune-inhibitory effect could be suppressed, and, therefore, an antitumoral response was promoted [58].

#### 3.1.4. Commonly Used Cellular Lines-Derived Evs

Other common cellular lines, used on a regular basis in the laboratory, can also be a source of EVs for drug delivery. These cell lines (such as human embryonic kidney 293 cell line (HEK293T), Chinese hamster ovary cell line (CHO) or the cervical cancer immortal (HeLa) cell line) are easy to be genetically manipulated and have been commonly used for protein modification and overexpression. For example, the cellular line HEK293T, is one of the most used cellular lines for research on EV-mediated drug delivery and shows potential for industrial applications. Although EVs derived from HEK293T cells can be enriched with some molecules from cancer-related pathways, HEK293T-derived EVs display high transfection efficiency and are easy to load with small therapeutic RNA molecules [65].

### 3.2. Modification and Loading of EVs

Different methods have been proven useful to upload therapeutic molecules into EVs. These therapeutic agents can either be chemotherapeutic agents or nucleic acids (RNA-based therapies) [34]. At the moment, there are two ways to create EVs containing a desired drug or molecule. It is possible to either load the drugs/molecules first into parental cells and then generate the release of EVs, which will already contain the given molecules, or incorporate the drugs/molecules into previously isolated EVs [1,43].

#### 3.2.1. Modification of Parental Cells

The engineering of parental cells, with the aim to transmit a determinate molecule to the EVs these cells secrete, can be performed through different methods. Subsequently, altered cells are cultured and secrete modified EVs containing the molecules of interest [66]. Of note, modifying the cells that will later produce the EVs may allow for designing exosomes to target specific tissues [10]. Moreover, using this methodology for cargo upload preserves the intact integrity of EVs membrane, which is usually damaged when other post-isolation loading techniques are employed [34]. Two of the perhaps most used approaches to engineer parental cells are the loading of these cells with exogenous cargo and the transfection of parental cells with DNA (Figure 3a) [1,56].

On one hand, the therapeutic of interest could be simply loaded in the parental cells from which exosomes will be isolated, through incubation of the cells with the drug [67]. For example, when a high dosage of PTX was cultured together with MSCs cell line SR4987 during 24 h, it was internalized by the cells and later released inside most EVs. Those EVs showed significant anticancer effect in vitro and in vivo when compared to a control group [62,68]. However, this method can cause cytotoxicity to the parental cell due to the drug loading and low efficacy of this drug loading within exosomes [1].

On the other hand, the DNA of interest could be transfected, and consequently alter and control the phenotype and cargo of the EVs derived from genetically modified cells [1]. Of note, not all cell lines are suitable for exogneneos expression, and the loading of EVs is difficult to control. In the study performed by O’Brien et al., the invasive triple negative breast cancer cell line (Hs578T) was engineered to overexpress miR-134. As a consequence, the EVs released by these cells contained the desired miRNA. Those were isolated and used with the aim to decrease the expression of Hsp90 in the cancer cell line. When the miR-134 entered via EVs to the targeted cells, cell migration has been reduced and the efficacy of the anti-Hsp90 treatments increased [1,69].

#### 3.2.2. Direct Loading of EVs

There is also the possibility to load functional therapeutic molecules, such as biomolecules, synthetic compounds or drugs, directly to previously isolated EVs (Figure 3b) [67,68]. Since the lipid bilayer represents in these cases a restriction for the loading, the different techniques must accomplish the final goal of bypassing the EV membrane without causing excessive damage [1]. They are usually referred to as active loading strategies, which can be chemically induced (with chemical agents such as transfection reagents or saponin) or physically induced (involving the disruption of the membrane with methodologies such as electroporation or sonication) [33,34]. Loaded EVs may be structurally modified and engineered to improve cancer therapy after loading, to enhance its homing abilities [70].

Depending on the nature of the cargo, different loading methods may be chosen, and, occasionally, the simple mixing of EVs with a free drug is enough [1,54]. For example, with some hydrophobic drugs (such as PTX), it is only required to mix the cargo with EVs to accomplish the loading and encapsulation in the vesicles. This allows for increasing drug solubility and stability. Some clinical trials have already used this methodology to deliver specific cargo to the tumor, for instance, curcumin-loaded EVs [58]. Free curcumin (an anti-inflammatory agent used for treating cancer) has been mixed with previously purified exosomes from a mouse tumor cell line (EL-4). The curcumin particles were successfully internalized by the exosomes, and those exosomes exerted positive effects when delivered to inflammatory cells, increasing the efficacy of the curcumin particles [68].

### 3.3. Evs Isolation Techniques

Once the best cellular line for a specific experiment has been chosen and, if required, parental cells have been engineered and loaded with the desired cargo, EVs (containing the drug or not yet modified) need to be isolated. Several EV isolation techniques can be used for this purpose. An optimal method is expected to demonstrate the high purity, high efficiency and high recovery yield of exosomes, as well as scalability and reproducibility [1,33].

Until date, several methods for EVs/exosome isolation have been described (Table 1).

#### 3.3.1. Differential Ultracentrifugation and Density Gradient Centrifugation

Differential ultracentrifugation (UC) is the current gold standard and most commonly used method for EV purification, as it is a cheap scalable technique and can be used in most circumstances. However, this method still presents some drawbacks. It is a low-yield, time-consuming method, difficult to automatize and with a high risk of EVs collapse or aggregation. Moreover, this process requires access to specialized instruments and training [33,41,71]. This isolation technique is based on sequential centrifugation for the sedimentation of EVs at high g-forces. It starts with low-speed spins to remove cells and large cellular debris. Later high-speed UC is used to pellet EVs. However, the resulting sample is usually contaminated with various types of EVs and protein aggregates. It is possible to further separate the different vesicle types by later sucrose density gradients to significantly improve the purity of the sample [1,33,40,43,72].

#### 3.3.2. Size-Based Filtration, Chromatography and Fractionation

Size-based filtration methods (i.e., tangential filtration, flow-field franctionation) together with chromatography-based separation are emergent large-scale EV isolation techniques, that are fast and automatable [33]. Nevertheless, the major weakness of this methodology is that the type of membrane used can have large impact on the quality of the isolated EVs [40]. With this methodology, EVs can be separated from the rest of the sample via sequential filtration using different filters with the desired pore size or molecular weight limit [1,72]. Heinemann et al. designed a three-step protocol with the aim to isolate EVs using only a filtration technique [33,73]. Initially, cell debris is being removed using a 0.1 μm pore size polyethersulfone (PES) membrane. Then, free proteins and a large volume of the sample are reduced using a 500 kDa molecular weight cut-off modified Polyethersulfone (PES) filter. The last step of the protocol consisted in the final EVs isolation with a 0.1 um Track Etch filter [33].

In addition, a chromatography method of special relevance for EVs isolation is size-exclusion chromatography (SEC). It is a promising technique as it allows for the separation of nanoscale particles depending on their hydrodynamic size [72]. It consists of a column filled with different-sized pores. Smaller-sized molecules will have to go through many pores, while larger molecules will be eluted faster [1]. SEC seems to present the high purity and integrity of the sample and advantages in different types of fluid, such as plasma or serum [40]. It is possible to combine SEC with UC for a better result [1,33,74].

Asymmetric Flow Field-Flow Fractionation is another used technique for EVs isolation based on their size. More commonly used for the separation of other types of nanoparticles, this methodology consists of the application of a laminar flow on the sample and a crossflow separation field which pushes the particulate molecules to an accumulation wall. Smaller particles will be reflected to the center of the chamber faster and eluted before larger ones. This technique has been reported to successfully isolate EVs sorted from a mouse melanoma cell line [1].

#### 3.3.3. Immunoaffinity

Immunoaffinity, is a method based on selective antibody-mediated arrest of EVs with specific surface proteins. Thanks to the specificity of antibodies receptors, this technique allows for a more selective isolation of exosomes. Specific antibodies are fixed on a surface of exosomes. Several washes are performed, consequently exosomes detached and are collected [1]. This technique allows one to obtain a higher purity of the sample, and the separation from the different subtypes of EVs could be performed [33,75]. It is used to isolate subpopulations of EVs derived from cell sources, such as cancer cells. For example, a method to specifically isolate exosomes derived from antigen-presenting cells used antibody-coated magnetic beads to capture a precise subtype of exosomes through the major histocompatibility complex class II [33]. However, protocols for immunoaffinity procedure are set on a very small scale and the costs for large volume samples isolation are highly expensive, which are important drawbacks for the clinical translation [33,40]. Moreover, it is hard to recover fully intact EVs [76].

#### 3.3.4. Polymer Precipitation

Another technique for EV isolation is a method based on polymer precipitation, commonly used to precipitate other molecules such as viruses. In this method, the sample containing the EVs is precipitated with a solution of polyethylene glycol (PEG). Then, with a centrifugation, it is possible to obtain a pellet containing EVs [1]. This approach for EVs isolation is easy, scalable and does not require long runs or specific equipment. Actually, different commercial EV isolation kits, such as Exosome Isolation (Thermo Fisher Scientific) or ExoQuick (System Biosciences), are used to simplify the EVs isolation process [40]. However, the purity of the sample obtained this way is not currently suitable for clinical application [40,77].

#### 3.3.5. Microfluidic Separation

Finally, microfluidics is a separation method widely used for other nanoparticles. This method is fast and presents high sensitivity as it can be combined with other techniques such as immunoaffinity methodologies, using surface protein markers to enrich the sample of exosomes without contamination [1,41,72]. This technique allows one to obtain relatively pure samples of EVs. For instance, CD41-positive platelet-derived EVs were isolated from plasma through a quick process using an anti-CD41 antibody-coated surface [33]. However, a strong limitation of this method is the low sample volume restriction [1,72].

### 3.4. EVs for Drug Delivery in the Clinics

The race to find the best type of EVs, isolation method and source for these potential drug delivery systems has already started. Currently, some clinical trials have by now demonstrated the promising application of EVs in the clinics (Table 2). Yet, these methodologies are at the early stages of investigation and clinical application [33]. It is, however, possible to find a few listed clinical trials already using EVs of different sources as drug delivery systems to treat cancer (Table 2) (http://clinicaltrials.gov [78]). Here, we highlight a study in phase I carried out in the USA (NCT03608631), expected to present results by March 2022, which aims to use mesenchymal stromal cell-derived EVs loaded with KrasG12D siRNA to fight against a specific type of pancreatic cancer. Another clinical trial in France (phase II), completed in 2018 (NCT01159288), assessed the potential of vaccinations with tumor antigen-loaded dendritic cell-derived exosomes against lung cancer, with the aim to activate the innate and adaptive immunity of the patients. Phase I of this study already showed safety of the treatment and feasibility [79]. However, the final results of its phase II have not yet been revealed.

To exemplify the use of non-human EVs sources for drug delivery, there is a phase I clinical study being carried out in the USA (NCT01294072), with the purpose of using plant-derived exosomes for the delivery of curcumin to targeted colon tumors, as previous clinical trials showed low efficiency and limited bioavailability of oral consumed curcumin, even in high doses [79]. This phase I study is scheduled to end by December 2022.

### 3.5. Artificial Extracellular Vesicles as Ideal Drug Delivery Systems

Each cellular type, loading method or isolation technique has certain potential for the production and obtention of efficient EVs for drug delivery. However, these qualities may not be enough on their own to generate an ideal EV-based DDS. Moreover, clinical use of this drug carriers is at the moment limited due to the low yield production of EVs by the different cell sources [80,81]. Yet, this knowledge could be used to synthesize artificial EVs specifically designed for drug delivery against a specific type of cancer. Several methodologies have already been described for the generation of artificial EVs, although more research and consensus among scientists in terms of biomaterials is needed [82].

Within the concept of artificial EVs as novel delivery systems, these can be separated into semi-synthetic EVs (which have been only modified before or after isolation) or fully synthetic EVs/EV mimetics (cell culture generated or artificial structures that mimic native EVs) [80,82]. These EV mimetic vehicles are usually produced on a large-scale by extrusion of specific cells (using micrometer-sized membranes) or built up from synthetic lipid materials forming liposomes [80,81]. Synthetic liposomes have been considered to be a viable vehicles for cancer therapy for a long time, as most cancers present a high number of light density lipoprotein receptors [83]. Therefore, liposomes or lipid-based nanoparticles have centered many efforts in the past few years. They were demonstrated to effectively load different drugs such as RNAs or chemotherapeutic agents. However, they were prone to present immunogenicity and toxicity as well as low ability to reach specific organs or tissues, which is a major drawback for human treatments [83]. The benefits of engineered vesicles that carry both the simplicity of liposomes, which can be easily modified, as well as specific EV membrane proteins, grant these artificial EVs the ideal characteristics for drug delivery [80,84].

Furthermore, artificial EVs can be engineered to present on their cell membrane surface other targeting ligands to improve their biodistribution and targeting capabilities. This method has been used as an example to create EVs expressing a fusion protein for the treatment of chronic myelogenous leukemia, a disease in which some patients develop drug resistance against common treatments (i.e., tyrosine kinase inhibitors) or have strong side effects because of inefficient site-specific accumulation of the drug [34,85]. In this case, HEK293T cells were used as the source of EVs, being first transfected with a plasmid containing the exosomal protein Lamp2b fused with interleukin 3 (IL-3). The IL-3 receptor is overexpressed in blasts from patients suffering chronic myelogenous leukemia; therefore, this molecule could be used for targeting purposes against this type of cancer. Moreover, EVs were loaded with Imantib, a first line leukemia treatment. In cell culture and mouse models, the efficacy of such EVs as drug delivery vehicles has been confirmed since the cytotoxicity of Imantib, when compared to EVs without the lamp2b-IL-3 fusion protein, was significantly reduced, while the survival rate augmented [85]. These results show that using targeting ligands can improve drug delivery to specific sites and this significantly improves the treatment.

Likewise, elements to avoid the activation of the immune system against the EVs could be incorporated into the vesicles to enhance their immune evasion properties. The most common approach in this context is the use of PEG (also used in polymer precipitation isolation technique), a molecule that forms a hydration layer around the vesicles which reduces their recognition by immune cells and therefore enhances the circulation time of the particles [34]. This was confirmed by a study where mice were injected with EVs from human epidermoid carcinoma cells fused with PEG. These vesicles could be found in blood after one hour, while non-PEGylated EVs had been completely cleared from circulation within ten minutes [34,86].

In addition, stimuli-responsive elements could be used to improve functionality and spatial action by adding, for example, peptides that are sensitive to the acid TME, such as pH-sensitive functional groups, generating the extracellular release of the drug only when the EVs are exposed to the acidic tumor environment [34]. As an example, EVs were modified with 3-(diethylamino) propylamine (DEAP), which causes the collapse of the EV membrane when the pH is below 7.0 [34,87]. Another pH-sensitive membrane functionalization approach which enhances EV uptake and cytosolic release is cationic lipid and pH-sensitive peptide (GALA) conjugation [10]. In both cases, the disruption of the membrane from EVs containing a drug allow the release of the drug to a targeted site [34]. However, additional types of stimuli-induced responses could also be useful to fight cancer and overcome the current failures (Figure 4).

## 4. Discussion

Efforts to develop new treatments based on nanomedicine applications have exponentially grown for the past decades with the final aim of improving the delivery of different treatments by using nanocarriers to a wide range of diseases, including cancer. This kind of therapy presents advantages when compared to conventional cancer therapy (i.e., Chemotherapy), in terms of improved solubility, enhanced circulation time, targeted delivery and reduction of adverse side effects. Yet, only a few synthetic DDS have reached the market so far, as only few of them have been completely safe and significantly improved patient outcomes [88,89]. One of the major drawbacks of synthetic nanoparticles is the insufficient accumulation of drug in the desired organ or tissue [88,90]. Indeed, despite many preclinical studies, only one synthetic nanoparticle with active targeting capacity is on the market today [91].

Many studies and reviews have discussed the possibility to modify EVs to target various diseases [1,34,38,54,56,66,67]. This interest arises from the specific role of EVs in cellular communication and their capacity to alter the recipient cell phenotype by transferring their inner content [33].

Possibly, the utmost benefit of EVs is their lack of immunogenicity [43]. Contrarily, synthetic nanoparticles such as liposomes may cause hypersensitivity reactions and immune rejection [65]. On the other hand, EVs present intrinsic targeting capabilities through ligands and receptors expressed on their membranes, which is a very important feature in order to achieve a targeted drug delivery system and might offer serious advantages in reduction of side effects and enhanced efficacy over other synthetic nanoparticles [43]. Synthetic nanoparticles have the tendency to become very fast opsonized with proteins in the blood stream, while the targeting features of EVs confer an important influence on bio-distribution of the drug, enhancing circulation time and cellular interactions with intrinsic homing abilities [55,58,92]. The complex composition of their surface membrane enables high specificity and selectivity for their targets [55]. Besides, nucleic acids (i.e., small RNA) might especially benefit from being delivered by EVs [58]. Although the benefits in using natural nanoparticles for drug delivery in cancer seem obvious, the mechanisms by which EVs are transported through the body and to their target cells or tissue are not yet fully understood. EV-based DDS need to be further studied and validated [34,93]. An additional problem of EVs-based DDS is the lack of high-throughput methods of isolation and efficient drug loading for clinical applications. Currently, most studies have been made under a small-scale EVs production protocol [33]. Yet, large-scale synthesis would be required for their clinical translation. Furthermore, it is also important to take into account the feasibility of manufacturing EV-based DDS under good manufacturing practices (GMP) [10]. On the other hand, synthetic DDS can be produced as a large-scale homogeneous population, with standardized protocols [34].

For EVs application as DDS, it is very important to consider the different features of the EVs. Some EVs, depending on their cell of origin, can stimulate immune and anti-tumor responses [58]. This illustrates that the choice of an appropriate cell type or cell state are essential questions for the production of efficient EV-based DDS in a given disease or therapeutic application [59,65]. As mentioned above, the most used sources for EVs are immune cells, MSCs, cancer cells or common cell lines. Immune cells used as an EVs source in clinical trials demonstrated that non-modified EVs are usually not enough to induce potent beneficial effects in vivo [59]. On the other hand, MSC-derived EVs do not affect the immune system [59]. Still, this kind of EV proved to reach targeted organs after infusion bypassing the lung microvasculature [94,95]. Confirmed differences exist in biological effects of MSC-derived EVs from different sources (bone marrow, adipose tissue or endothelium) [94]. Although this cell source seems to be the most extended in use, its application in clinical trials is still limited, as some critical parameters, such as culture conditions or protocols for production, storage or administration, are not standardized [94,95,96]. On the other hand, the use of cancer cell lines as a source of EVs may simplify their isolation. However, the yet unknown content shared through cancer cells with EVs may represent an important safety risk [58]. Finally, EVs obtained from common cellular lines may be easier to produce in large quantities, although with fewer biological benefits. Other studies point to using EVs of vegetable origin (from freshly prepared juice of edible plants) as a source for drug delivery. For example, EVs derived from grapefruit juice proved to successfully deliver interfering RNA, proteins or chemotherapeutic agents in animal models [58]. Moreover, other non-human sources of EVs are being tested, like EVs derived from animal milk, which proved to successfully function as drug carriers. Attention should be paid to this particular source of EVs as it might allow the production of low-budgeted and up- scaled EVs [58,97].

Further, the need to find efficient isolation methods for EVs is one of the major limitations to use EVs for drug delivery [40,63]. Conventional isolation techniques usually present low purity of the sample and limited recovery yields [34]. Moreover, standardized protocols for the purification and isolation processes are also needed. These protocols should be scalable to translate the techniques into large-scale EVs production under GMP [33,43]. Yet, it is important to bear in mind that the combination of two or more methods may improve the isolation of EVs and the scalability of the process [1]. Likewise, with the current isolation methods, it was formerly impossible to completely isolate pure exosome samples [59]. Until date, the most used EVs isolation techniques have been UC combined with density gradient centrifugation, polymer precipitation or SEC for further purification and possible large-scale production of EVs [40,58,63,74]. Lately, novel technologies to isolate EVs are being developed. Microfluidic platforms are an example of such technologies; they have great potential but still need to be optimized in order to standardize the protocols and storage conditions, to maintain the functionality of EVs and make large-scale isolation feasible [58,63,72].

As synthetic nanoparticles have so far failed in their translation to the clinics, and the development of EVs for drug delivery is still facing some major challenging issues for large-scale production, artificial extracellular vesicles may represent an ideal DDS connecting the best of both systems [98]. These carriers mimic the structure of EVs, although conserving the simple structure and characteristics of synthetic DDS [80]. Additionally, large-scale production would be easily achieved in a short time period, and vesicle loading would become a simpler process [80,81,99]. Some in-between alternatives for synthetic nanoparticles and EVs have already been investigated, such as cell-membrane-coated nanoparticles. They use natural membranes and therefore benefit from their biological characteristics as EVs [34]. Cell-membrane-coated nanoparticles thus carry both properties of synthetic nanoparticles and cellular membranes. As a drawback, the extraction of this cell membrane is a complicated and time-consuming process [34,100].

## 5. Conclusions

Altogether, EVs are prospective for cancer treatment; however, their functionality and physiological role are still under investigation [40,63]. Moreover, stronger preclinical models (immunocompetent mice, humanized patient derived xenograft (PDX) models) to predict the human response to the treatment are needed for early phase clinical trials [58]. Additionally, a better understanding of EV biological function is needed, and several issues related to the purification, loading, targeting and scaling-up of EVs as DDS must be carefully considered to successfully transit these particles from bench to bedside [40,63,94].

## Figures and Tables

**Figure 1 pharmaceutics-12-01146-f001:**
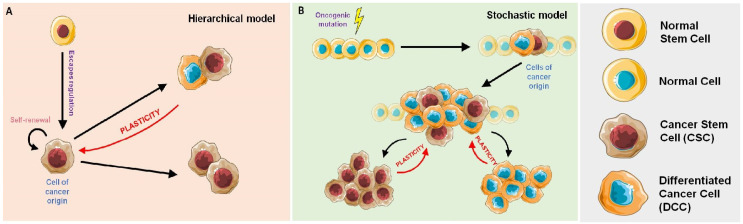
**Tumor cell models**. (**A**) Hierarchical model of division: a cancer stem cell (CSC) is originated from a normal stem cell that escapes from cell cycle regulation. This CSC has self-renewal capacity and acts as the cell of origin of the tumor and can generate different types of tumor cells. Because of plasticity, those differentiated cells can reverse their phenotype into CSCs. (**B**) Stochastic model of division: The cells of origin of the tumor can be any type of cell that experiences oncogenic mutations. Some mutations can lead to stem-like phenotypes and thus the cells become CSCs. This phenomenon (plasticity) unites the hierarchical model with the stochastic model.

**Figure 2 pharmaceutics-12-01146-f002:**
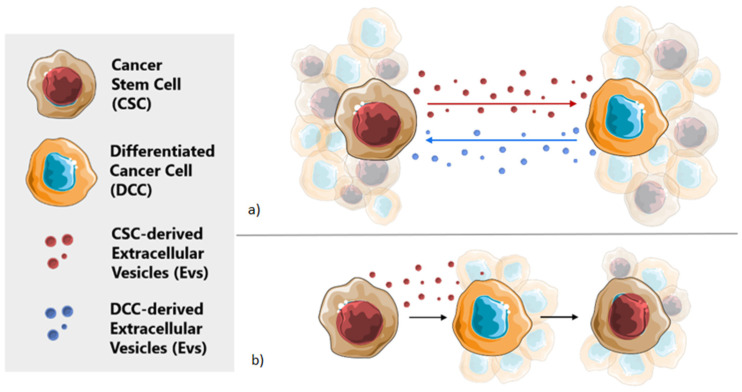
**Cellular crosstalk with Extracellular vesicles** (**EVs**). (**a**) EV-mediated molecular crosstalk between cancer stem cells (CSCs) and differentiated cancer cells (DCCs) allows for maintaining stable subpopulations. (**b**) Thanks to this cellular communication, DCCs can reverse to the CSCs phenotype when the CSC population decreases and vice versa.

**Figure 3 pharmaceutics-12-01146-f003:**
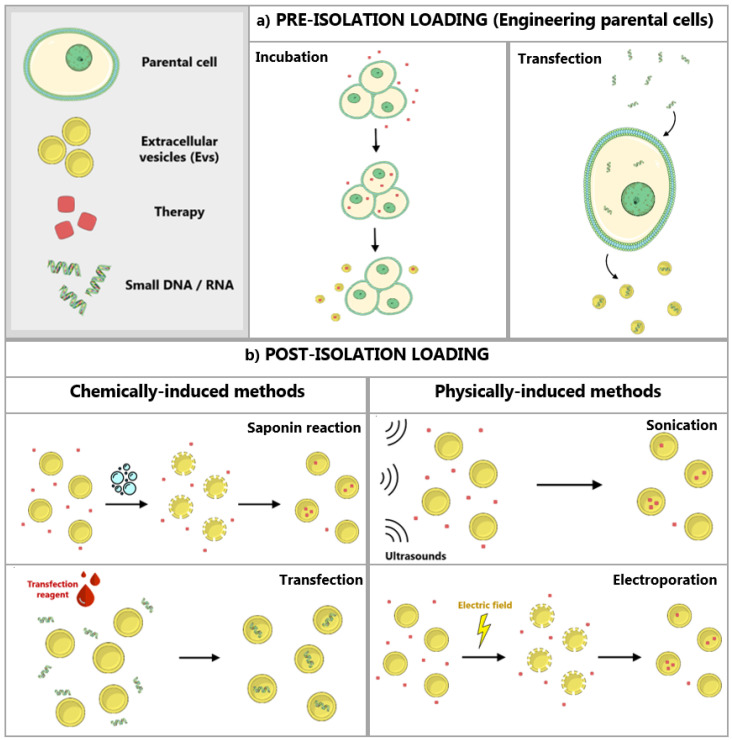
**Extracellular vesicles** (**EVs**) **loading methods**. (**a**) EVs can be loaded before isolation by engineering their parental cells. This procedure can be achieved by incubation of the parental cells with the desired cargo or by transfection of the cells. EVs secreted by those cells will already contain the cargo. (**b**) EVs can be loaded with a desired cargo after being isolated from the sample. This procedure can be chemically induced (through transfection or saponin reaction) or physically induced (through sonication or electroporation).

**Figure 4 pharmaceutics-12-01146-f004:**
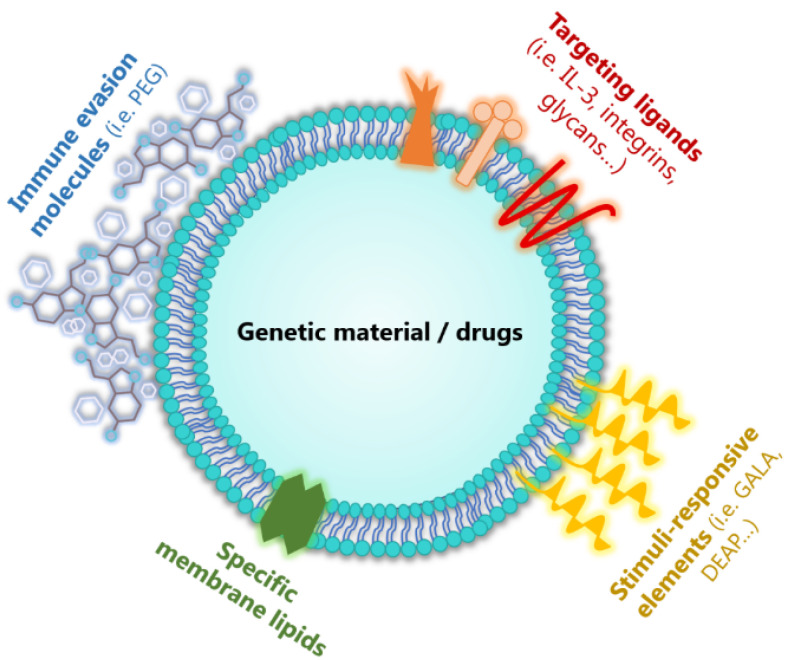
**Structure of an ideal extracellular vesicle** (**EV**) **for drug delivery**. EVs can be artificially synthetized or engineered to gain potential as ideal Drug Delivery Systems (DDS). While maintaining their natural membrane or synthetizing a simple liposome, immunosuppressive molecules (such as polyethylene glycol (PEG)) could be added to the membrane to avoid the action of the immune system of the patient. Moreover, different targeted ligands (like IL-3, integrins or glycans) could be used to direct the vesicles with the therapy to specific cells or tissues delivery. Moreover, stimuli-responsive elements (for instance pH-sensitive peptide (GALA) or 3-(diethylamino) propylamine (DEAP)) help to deliver their cargo with more specificity. Within the inner core, EVs may contain genetic material (like siRNA for therapy) of drugs against cancer or other diseases.

**Table 1 pharmaceutics-12-01146-t001:** Extracellular vesicle (EV) isolation techniques.

Isolation Method	Procedure	Advantages	Disadvantages
Differential Ultracentrifugation (UC)	The different molecules in a fluid sample are separated by centrifugation at high g-forces. Can be combined with sucrose density gradients or SEC for higher purity.	As the gold standard for EVs isolation, it is a cheap and scalable technique.	Low-yield technique with a time-consuming protocol, difficult to automatize. Moreover, specialized instruments and training are needed. EVs may collapse and the resulting sample is usually contaminated.
Size-Based Filtration, Chromatography and Fractionation	Technique based on a column filled with different sized pores. Smaller size molecules will have to go through many pores while larger molecules will be faster eluted.	Fast (normally a single step) and automatable method with high purity and integrity of the resulting sample.	The type of membrane used can have a large impact on the quality of the isolated EVs.
Immunoaffinity	Selective antibody-mediated arrest of EVs with specific surface antigens.	Allows a more selective isolation of EVs.	Protocols for immunoaffinity procedure are set on a very small scale and the costs for large volume samples isolation are high. Also, it is hard to recover fully intact EVs.
Polymer precipitation	The sample containing the EVs is precipitated with a solution of PEG and concentrated by centrifugation	Easy, scalable technique that does not require long runs or specific equipment.	The purity of the sample obtained should be improved. It is frequent to have samples contaminated with other particles and proteins.
Microfluidic separation	This method uses different techniques like immunoaffinity or filtrations to isolate EVs.	Fast technique with high sensitivity and efficiency.	This method presents a low sample volume restriction and needs expensive devices.

**Table 2 pharmaceutics-12-01146-t002:** Clinical trials with EVs as drug delivery systems.

Type of Cancer	EV Source	Isolation Method	Loading Method	Therapeutic Cargo	Phase	Ref.
Malignant pleural effusion	Tumor cells	Not mentioned	Not mentioned	Chemotherapy	Phase II	NCT01854866
Non-small cell lung cancer	Dendritic cells	Ultrafiltration/UC	Not mentioned	Peptides	Phase II	NCT01159288
Pancreatic cancer	Mesenchymal stromal cells	Not mentioned	Not mentioned	KrasG12D siRNA	Phase I	NCT03608631
Melanoma (stage III/IV)	Autologous monocyte-derived dendritic cells	Ultrafiltration/UC	Incubation with parental cells	MAGE3	Phase I	[78]
Lung cancer (stage IV)	Human lung carcinoma cell line A549	Differential gradient centrifugation	Passive incubation	Cisplatin	Phase I	[64]
Colon cancer	Plant nanovesicles	Not mentioned	Not mentioned	Curcumin	Phase I	NCT01294072

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
