# Peer review of "Extracellular Vesicles as Drug Delivery Systems in Cancer"

_pharmaceutics, 2020, doi:10.3390/pharmaceutics12121146_

Round 1

Reviewer 1 Report

The present article is an overview on the field, as many others

Although it is a well-written and organized paper nothing new is added is the already published literature on the field

For example, no reference on the immediate potential clinical application of this method and no important clinical trials are under way. Still remais a theoretical topic

Author Response

We thank reviewer 1 for her/his comments and for stating that our review is a well-written and organized paper. We agree that still no important clinical trials are under way regarding the use of EV against cancer indications but this should be considered in regards of the fact that the field is still very on its early steps towards this goal. Besides the many challenges that the field still has ahead to succeed on this, as reflected in our review, many efforts are being done worldwide and new promising data is constantly published.

Reviewer 2 Report

The scope of this review is to summarize recent knowledge regarding the use of EVs to treat cancer stem cells and cancer resistance.

The work is fairly well written but needs a lot of improvement in terms of division into chapters and general organization.

Major comments:

  1. Chapter one introduces the problem of cancer and related issues: it is well written although could be shortened highlighting only the aspects that could be addressed by therapy. Authors could also introduce the scheme of the review at this point.
  2. Chapter 2 is not needed in this review, it leads to confusion. In this review authors are focusing on drug delivery properties of EVs and they are not describing the properties of EVs in cancer, this is not the focus. Indeed later on they talk about EVs derived from other cells (immune cells and so on). So in the reviewer opinion, this chapter should be removed: authors could cite the importance of EVs in the first chapter but this is not even important since then they do not talk about specific drugs that limit EV secretion or similar. EV description and general properties may be written in chapter 3.
  3. I would suggest to put here, before chapter 3, an introduction on specific drug delivery approaches other than EVs used in the cancer field, to better introduce the need of using EVs in this field. This topic is only discussed in the discussion section but the reviewer believes should be introduced at this point (use of liposomes, synthetic nanoparticles).
  4. What is the difference between “modified” EVs described in paragraph 3.2 and “artificial” EVs described in 3.5? It is not clear why paragraph 3.5 should repeat the same concepts (modification of EVs by incorporation of nucleic acids for example). Furthermore paragraph 3.5 comes after 3.4 in which authors describe the clinical use of EVs, referring also to modified EVs.
  5. The discussion section is tremendously long and repetitive of the same concepts described in the previous paragraphs. It should be better to discuss issues regarding the use of EVs in each specific chapter. In this way it would be easier to find a topic of interests with pros and cons. Maybe authors should end with a discussion and conclusion chapter, not repeating the concepts but highlighting the most important aspects to take into consideration in the field and future perspectives.

Minor comments:

Figure 1: please use bigger fonts.

Line 197: please explain acronym “PK/PD”

Line 214:the sentence “EVs looks a lot like liposomes” is not very scientific. Could it be rephrased?

Line 192: substitute “gather” with “gathered”

Line 248: Delete “Briefly,” at the end of the sentence. Or a sentence is missing.

Line 257: substitute “trough” with “through”

Line 300: substitute “ manipulate” with “manipulated”

Line 308: rephrase this sentence “can either be chemotherapy or nucleic acids”. Either refer to the agent or to the therapy in both cases.

Line 335: what are the drawbacks of this technique?

Figure 3: put the (a) close to the word “Pre-isolation…”

Line 361 and others: delete “s” from “EVs”. Sentence should become “EV isolated techniques”. Please check this in the manuscript.

Line 382: Reviewer does not agree with the sentence “low rate of recovery”. Indeed in the work of Busatto et al. they show a very high recovery rate with TFF.( doi: 10.3390/cells7120273 )

Paragraph 3.3.4: the concept of purity needs to be  better explained by the authors. Using polymer precipitation from a plasma sample determines the isolation of many nanoparticles . See Salvi et al. 10.3892/ijmm.2019.4158. Furthermore, being the secretome nanostructured, we expect to separate many different nanoparticles by this technique ( see Busatto, S., et al. (2019). "The nanostructured secretome." Biomater Sci 8(1): 39-63.).

Table 1: Flow field techniques are not cited in the table. Please revise.

Line 525: put “grown” instead of “Grow”

Author Response

Dear reviewer,

Thank you for all the comments and suggestion. We addressed them carefully. Our replies are as follow:

  1. Chapter one introduces the problem of cancer and related issues: it is well written although could be shortened highlighting only the aspects that could be addressed by therapy. Authors could also introduce the scheme of the review at this point.

As suggested the introduction part has been shortened substantially and a brief scheme of the review has been introduced.

  1. Chapter 2 is not needed in this review, it leads to confusion. In this review authors are focusing on drug delivery properties of EVs and they are not describing the properties of EVs in cancer, this is not the focus. Indeed later on they talk about EVs derived from other cells (immune cells and so on). So in the reviewer opinion, this chapter should be removed: authors could cite the importance of EVs in the first chapter but this is not even important since then they do not talk about specific drugs that limit EV secretion or similar. EV description and general properties may be written in chapter 3.

Thank you for this comment, we had carefully considered to remove the chapter 2, nevertheless we believe that it is important to describe the role of EVs within the tumoral niche in order to explained why it would be possible to use the vesicles as drug delivery for cancer treatment. Since the communication within different TME players is not completely understood its description may result confusing. Hopefully soon there will be more knowledge in respect and more light will be shed into the topic. Of note, we just publish an extensive review in this regard in Nanomedicine journal.

  1. I would suggest to put here, before chapter 3, an introduction on specific drug delivery approaches other than EVs used in the cancer field, to better introduce the need of using EVs in this field. This topic is only discussed in the discussion section but the reviewer believes should be introduced at this point (use of liposomes, synthetic nanoparticles).

Thank you for this suggestion. Accordingly we have introduced the use of nanomedicine in the beginning of chapter 3.

  1. What is the difference between “modified” EVs described in paragraph 3.2 and “artificial” EVs described in 3.5? It is not clear why paragraph 3.5 should repeat the same concepts (modification of EVs by incorporation of nucleic acids for example). Furthermore paragraph 3.5 comes after 3.4 in which authors describe the clinical use of EVs, referring also to modified EVs.

Dear reviewer, the chapter 3.5. stays alone and after the chapter 3.4. since it is description of artificial (synthetic) extracellular vesicles for drug delivery that is still quite futuristic and has not been used in clinical trials yet. We wanted to describe the possible future development of the field and thus this part is in the end of the manuscript, just before the discussion. Contrary, modified vesicles are natural extracellular vesicles modified by gene engineering.

  1. The discussion section is tremendously long and repetitive of the same concepts described in the previous paragraphs. It should be better to discuss issues regarding the use of EVs in each specific chapter. In this way it would be easier to find a topic of interests with pros and cons. Maybe authors should end with a discussion and conclusion chapter, not repeating the concepts but highlighting the most important aspects to take into consideration in the field and future perspectives.

Thank you for this comment. Accordingly, we have shortened significantly the discussion as well conclusion part of the manuscript.

In addition all minor comments have been carefully revised and manuscript has been edited accordingly. An additional grammar check has been provided. We believe that after those corrections the manuscript ameliorated substantially.

Thank you again for all the comments.

The authors

Reviewer 3 Report

The only somewhat "fake news" of this submitted manuscript is the sentence: " EVs were recently referred as " Trojan_horses_” for drug_delivery " (line 236)!. In fact, "The Trojan exosome hypothesis" is not a recent concept. It was first suggested in 2003 by Stephen J. Gould, Amy M. Booth, and James E. K. Hildreth. PNAS 16, 2003 100 (19) 10592-10597; https://doi.org/10.1073/pnas.1831413100. At about the same time, a given type of EVs was already patented as interesting vectors for theranostic drug delivery in human cells. However the idea had still to wait many years for the necessary establishment of a convenient environment about EV knowledge.

Coming back to your manuscript, this review is indeed quite interesting, well structured and well documented. The assets of EVs, either natural or engineered, are nicely advocated by comparison with liposomes or synthetic drug delivery particles.The review covers mostly in vitro studies taking into account the main cells currently used as EV sources; it is also concerned with some in vivo studies, and even with the few ongoing (phase I or II) clinical studies. Moreover, you are well aware of the many challenges, which remain to be solved in order to translate the worldwide increasing current knowledge about EVs from the bench to the theranostics of human cancers.

Briefly, I really enjoyed reviewing your manuscript, and suggest only some minor modifications for a slight improvement of the joined reviewed pdf.

Author Response

Dear Reviewer,

Thank you very much for your kind and detailed revision of our article. All the minor changes you have suggested have been carefully addressed and the manuscript text have been changed and /corrected accordingly. Besides, all spotted ambiguous statement were deleted or replaced. We believe that these changes have ameliorated quality and coherence of the manuscript.

Thank you again for all the comments.

Sincerely

The authors

Round 2

Reviewer 1 Report

The revised manuscript is much better

Recognizing the updated interest in the filed I suggest acceptance for publication